# Recent Advances in Graphene-Enabled Silicon-Based High-Speed Optoelectronic Devices—A Review

Yadvendra Singh  and Harish Subbaraman *

School of Electrical Engineering and Computer Science, Oregon State University, Corvallis, OR 97331, USA; singhy@oregonstate.edu
* Correspondence: harish.subbaraman@oregonstate.edu

**Abstract:** Silicon (Si) photonics has emerged as a prominent technology for meeting the escalating requirements of high-speed data transmission in optical communication systems. These systems need to be compact, energy-efficient, and capable of handling large amounts of data, driven by the advent of next-generation communication devices. Recently, there have been significant activities in exploring graphene within silicon-based components to enhance the overall performance metrics of optoelectronic subsystems. Graphene's high mobility of charge carriers makes it appealing for the next generation of high-performance devices, especially in high-speed optoelectronics. However, due to its zero bandgap, graphene is unlikely to replace silicon entirely, but it exhibits potential as a catalyst for silicon-based devices, namely in high-speed electronics and optical modulators, where its distinctive characteristics can facilitate progress in silicon photonics and other fields. This paper aims to provide an objective review of the advances made within the realm of graphene-integrated Si photonics for high-speed light modulation and detection.

**Keywords:** silicon photonics; electro-optic modulators; thermo-optic modulators; photodetectors



## 1. Introduction

The rapid expansion of data storage capabilities has created a need for high-speed transceivers to facilitate effective information transfer. As a result, communication devices are now required to include the quickest and most efficient components inside their integrated platforms [1–3]. With its low cost and strong compatibility with the complementary metal-oxide-semiconductor (CMOS) industry, silicon photonics has great promise for bridging the information gap between the electrical and optical domains in large-scale silicon photonic integrated circuits (PICs) of the future [1]. Nevertheless, the generation, modulation, and detection of light in a monolithically integrated silicon chip pose significant challenges for traditional silicon photonic integrated circuits (PICs). Despite the potential for epitaxial growth of III-V materials on silicon chips for light generation and detection, a challenge exists due to the lattice-constant mismatch and thermal expansion coefficient at the interface between silicon and III-V materials [4,5]. The interface defects also limit the ultimate devices' optical and electrical performance. Moreover, due to the limited performance of the pure silicon devices due to their carrier drift velocity (1000 cm$^2$/(V·s)) and indirect bandgap (1.14 eV), it is difficult to achieve extraordinary performance and miniaturization even from such silicon–III-V hybrid devices [6].

Since its initial discovery in the early 2000s, 2D materials have gained significant attention due to their extraordinary electrical and optical properties [7–10]. Among these 2D materials, graphene has several interesting properties, such as ultra-high electron mobility, mechanical strength, and high transparency from 1400 nm to 1600 nm in the near-infrared spectrum [7,11,12]. However, it exhibits a zero-band gap, which precludes its use as a semiconducting material in device applications [13]. However, its exceptional electrical conductivity and tunable optical characteristics make it an ideal candidate for enhancing the performance of silicon-based photonic devices. Integrating graphene onto silicon

waveguides has enabled a wide range of applications, such as high-speed data transmission [14], optical modulation and detection [12,15–19], sensors [20,21], and other photonic technologies [22–29], offering advantages such as low power consumption, compact size, and tunability. Here, we discuss the recent advances made in the field of graphene-based silicon modulators and photodetectors.

## 2. High-Speed Graphene-Enabled Silicon Modulators

In optics and photonics, an optical modulator refers to a device utilized to manipulate or modulate various characteristics of light, such as its intensity, phase, or polarization [30,31]. This modulation process is employed to encode information onto an optical carrier signal within short- and long-distance communication links. Optical modulators play a crucial role in a wide range of optical communication and information processing systems.

When integrated into silicon-based photonic devices to either tune absorption or phase within a Mach–Zehnder or Ring Resonator configuration, graphene has been shown to significantly improve key performance metrics, particularly in terms of bandwidth and power consumption. This means that by harnessing the unique properties of graphene, it becomes possible to achieve higher data transfer rates (bandwidth) and reduce the energy required (power consumption) in the design of efficient optical communication systems, making it a valuable asset in the ongoing optimization of such systems. Hence, we discuss the latest developments in graphene-based electro-optic and thermo-optic modulation devices in the next subsections.

### 2.1. Graphene-Based Electro-Optic Modulation Devices

Electro-optic silicon/graphene modulators, capitalizing on graphene's exceptional carrier mobility, present an exciting avenue for pushing modulation speeds to new frontiers, potentially reaching speeds of hundreds of gigahertz [32]. This makes them highly attractive and promising for advancing data and telecommunication systems, ensuring they can keep pace with the relentless demands of the digital age.

In electro-optic modulators, modulation depth is an important metric that determines their efficacy. Several researchers have proposed various techniques to increase the modulation depth by increasing absorption within graphene layers [16–18,30,33]. It was first achieved by Liu et al., who used two graphene layers separated by an oxide layer to create a p-i-n-like junction and demonstrated exceptional performance, achieving an operational frequency of 1 GHz [34]. This device design excels with a substantial modulation depth of approximately 0.16 dB/μm, all while requiring a reasonably moderate drive voltage of ~5 V. This innovative design harnesses the symmetrical band structure of graphene near the Dirac point, effectively mitigating the prevalent optical losses encountered in silicon photonics. Consequently, it offers numerous advantages, including a compact form factor, minimal energy consumption, and negligible insertion loss, as shown in Figure 1a [35]. While significant studies [30,34,35] have been made in optimizing modulation depths within the devices, as mentioned above, a notable constraint remains in the form of limited modulation bandwidth, typically capped at less than 1 GHz. This limitation predominantly arises from the substantial RC (resistance–capacitance) constant inherent to the device's circuitry, which constrains the speed at which modulation can effectively occur. Overcoming this bottleneck and achieving higher bandwidths, Phare et al. rigorously worked on the micro-ring resonator (MRR) structures [17] and, for the first time, demonstrated an electro-optic graphene modulator by integrating graphene over a ring resonator fabricated from low-temperature plasma-enhanced chemical-vapor-deposited (PECVD) silicon nitride [17]. The modulation efficiency of this device was 15 dB per 10 V and operated with 30 GHz bandwidth. Shortly afterwards, the same team also demonstrated a graphene-based electro-optic modulator with a bandwidth of 200 GHz at 4.9 K [36]. This device showed high performance at cryogenic temperatures. Typically, interference-based silicon devices such as resonators or Mach−Zehnders are highly sensitive to high-temperature operation due to the large

thermo-optic coefficient effect in silicon. Hence, to study the temperature-dependence of the operating characteristics, Dalir et al. experimentally demonstrated an athermal graphene optical modulator that exhibits a wide bandwidth of 140 nm throughout a wide optical communication range (1500–1640 nm). This device was based on the Si-waveguide planer structure using a double-layer graphene configuration, as shown in Figure 1b, which enabled it to achieve a modulation speed of 35 GHz with a power consumption of 1.4 pJ/bit, and 25 V was required to turn the modulator from an OFF to ON state [37]. Recently, in 2021, Agarwal et al. demonstrated an ultrafast electro-absorption modulator based on the integration of hexagonal boron nitride (hBN) and a high k-dielectric material hafnium oxide ($HfO_2$), as shown in Figure 1c [18]. This device achieved a ~39 GHz bandwidth with ER ~4.4 dB and IL ~7.8 dB due to an overcautious $V_{BT}$ = 12.1 V applied voltage. Within the realm of graphene-based electro-absorption modulators, Amin et al. reported a modulator for neuromorphic nonlinear activation by using an ITO-graphene heterojunction, as shown in Figure 1d [38]. Their device demonstrated a high modulation depth of 0.132 dB/μm, a 3 dB bandwidth of >130 GHz with a drive voltage of 20 V, and an insertion loss of <2 dB. Considerable research efforts have been dedicated to investigating Si photonic-graphene materials' heterostructures for the development of electro-optic modulators. Table 1 shows the relevant information about these devices' essential characteristics.

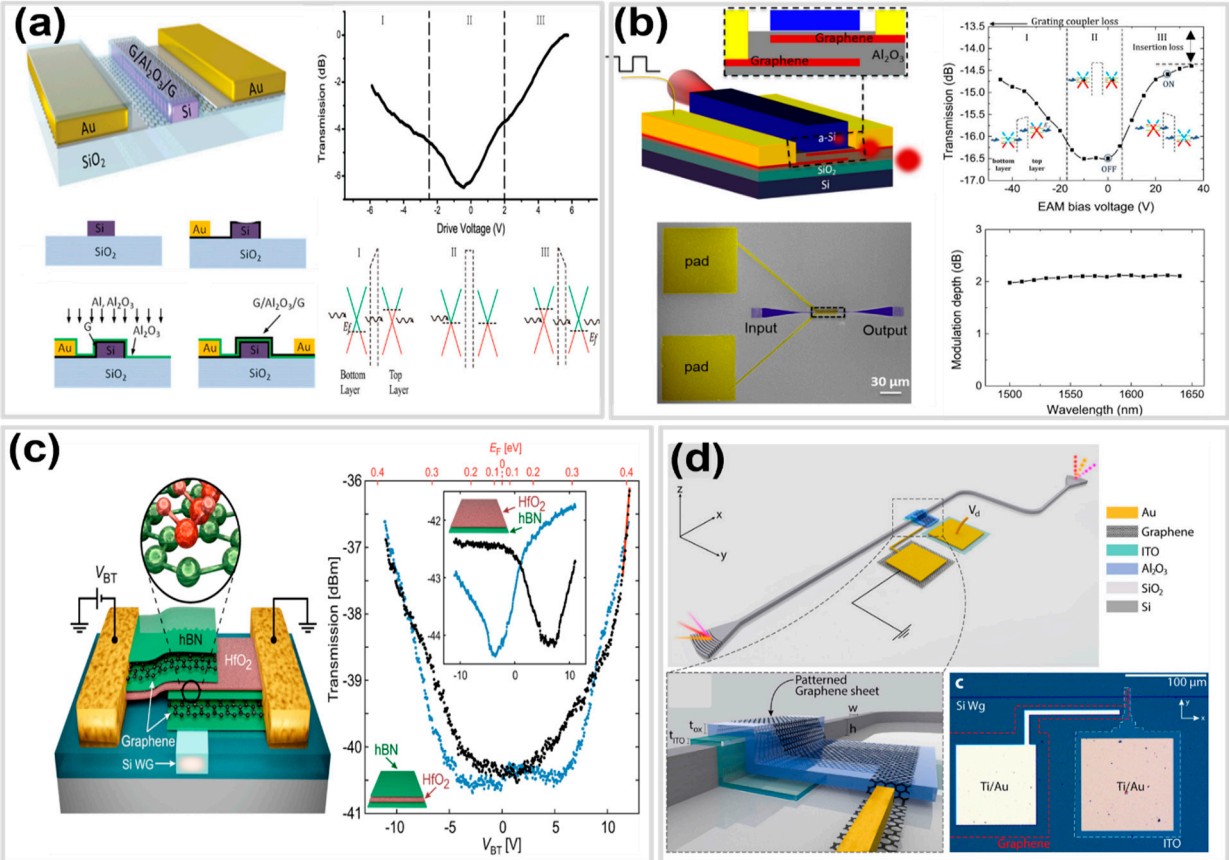

**Figure 1.** Graphene-assisted electro-optic modulators (**a**) Double-layer graphene modulator with static response and modulation depth of ~6.5 dB [35]. (**b**) High-speed broadband modulator with modulation depth of ~2 dB [37]. (**c**) hBN–$HfO_2$–hBN-based graphene EA modulator with transmission curves as a function of the voltage between the bottom and top graphene electrodes [18], and (**d**) ITO-graphene heterojunction absorption modulator [38].

**Table 1.** Performance summary of graphene-based electro-optical modulators.

| Year | Structure | Wavelength [μm] | Modulation Depth [dB] | Size [μm²] | Bandwidth [GHz] | Drive Voltage | Power Consumption [fj/bit] | Ref. |
|------|-----------|-----------------|----------------------|------------|-----------------|---------------|---------------------------|------|
| 2011 | Si straight WG | 1.35–1.6 | 2.3 | 25 | 1.2 | 4 | 1200 | [34] |
| 2012 | Si straight WG | 1.55/3.5 | 4.77 | 63 | 120/30 | 8/3.5 | - | [39] |
| 2013 | Si PhCC | 1.57 | 10 | 108 | 300 | 1.5 | 340 | [40] |
| 2014 | Si straight WG | 1.55 | 7.042 | 225 | 2.5 | −40 | - | [41] |
| 2014 | Si MRR | 1.55 | 3.68 | 141 | - | 6 | - | [42] |
| 2015 | Si 1D grating | 1.56 | >10 | 54 | 45 | 4.8 | - | [43] |
| 2015 | Si MRR | 1.555 | 12.5 | 127 | - | 8.8 | - | [44] |
| 2015 | Si3N4 MRR | 1.57 | 15 | 1680 | 30 | 10 | 800 | [17] |
| 2016 | Si straight WG | 1.53–1.565 | 12.79 | 500 | 2.6–5.9 | 2.5 | 350 | [45] |
| 2016 | Si PhCC | 1.55 | 3.2 | 100 | 1.2 | 2.5 | 1000 | [46] |
| 2016 | Si MZI | 1.55 | 3 | - | - | 8.9 | - | [47] |
| 2018 | Si straight WG | - | 1.5 | - | 100 | - | 15 | [48] |
| 2021 | Si straight WG | - | 12 | 27 | 39 | 10.4 | - | [18] |

### 2.2. Graphene-Based Thermo-Optic Modulation Devices

In photonics, thermo-optic modulation is an alternative way to modulate light data. The temperature-dependent refractive index of materials, which changes as the temperature changes, is used in this method. When you change a material's temperature, its refractive index changes. This causes changes in the phase and length of the optical path that light travels through. In the conventional silicon-based thermo-optic modulators, this modulation technique is often used by using a metallic microheater that is strategically placed near the silicon waveguide [33,49,50] or by integrating a conducting p-i-n structure closely with, or as an integral component of, the optical waveguide [51]. An important factor to take into account is the integration of a thin silicon dioxide layer, about 1 μm in thickness, between the metallic microheater and the waveguide. The primary purpose of this layer is to restrict the heating effect to the waveguide area, hence reducing the significant light absorption that is intrinsic to the metallic microheater. Nevertheless, the existence of this insulating layer presents difficulties concerning the speed and effectiveness of thermal energy transmission from the heater to the waveguide. The presence of this bottleneck results in increased power consumption and limits the achievable modulation speed. In the context of p-i-n based structures, the incorporation of carriers during modulation leads to supplementary optical losses via photonic absorption, specifically caused by mid-band donor states. The solution to these difficulties is crucial for improving the overall effectiveness and productivity of thermo-optic phase modulators in integrated photonic systems. This feature is very useful for photonic systems that need to control and modulate light. While studying silicon (Si) photonics, thermo-optic modulation has gotten a lot of attention and has been used a lot. Silicon has been used for a long time and is known for having a high thermo-optic coefficient, which means it reacts very quickly to changes in temperature. Different heat optical devices in Si photonic systems are based on this feature. As of late, adding two-dimensional (2D) materials to thermo-optic modulators, like graphene, has made new options possible. Because of its unique qualities, graphene has become a great choice for thermal devices. Its unique ability to absorb light efficiently across a wide range of spectrums makes it stand out. Graphene also has a high thermal conductivity, which means it can quickly shed heat. This makes it perfect for use in thermal uses.

Thermo-optic modulators based on graphene typically employ a waveguide structure, where graphene is integrated as a heating element. The operation relies on the thermo-optic effect, where changes in temperature induce variations in the refractive index of the waveguide material, leading to phase and intensity modulation of the guided optical signal [33,49,50]. Graphene's excellent thermal properties ensure efficient heat generation and rapid thermal response [52], allowing for high-speed modulation with silicon devices. In 2014, Yu et al. demonstrated the first application of graphene micro-heating in silicon photonics and proposed the MZI structure-based thermo-optic modulator as shown in Figure 2a [53]. The calculated heating efficiency of the device was 11.6 K·$\mu$m$^3$/mW as they increased the thermal conductivity of graphene by 5000 W/mK. It was further enhanced from 11.6 K·$\mu$m$^3$/mW to 35.5 K·$\mu$m$^3$/mW, and decaying time was reduced to 9 $\mu$s by reducing the gap between the metal pad and Si waveguide. Later, Gan et al. proposed a graphene-assisted MRR-based thermo-optic modulator [54]. They used the electrical and thermal properties of the graphene to enhance the performance of the device and achieved a high modulation depth of 7 dB with a tuning efficiency of 0.1 nm/mW. This structure was adapted by other researchers to increase the performance of the MRRs and micro-disk resonator (MDR)-based thermo-optic modulators [55–58]. In these graphene-assisted thermo-optic silicon modulators, a precise patterning technique is employed to selectively cover only the optical mode region with graphene, as opposed to the complete coverage of the entire device. By restricting the graphene coverage to the optical mode area, the modulator's performance can be fine-tuned, enhancing its functionality in a targeted manner and effectively leveraging the inherent properties of graphene for tailored modulation capabilities [55,56]. Yu et al. experimentally demonstrated that transparent graphene nanoheaters are also a good option for other thermally tunable photonic integrated devices, as shown in Figure 2c [56]. Other than the MRR and MDR structures, Yan et al. 2017 experimentally demonstrated an energy-efficient graphene microheater on a photonic crystal waveguide with a tuning efficiency of 1.07 nm/mW [57]. The fastest rise and fall times they demonstrated were 750 ns and 525 ns, respectively. Xu et al. proposed a graphene micro-heater-based tunable silicon nanobeam cavity with a very small optical mode volume of 0.145 $\mu$m$^3$ [58]. Because of this small optical mode volume, graphene interacted with light properly and achieved a thermo-optic tuning efficiency of 3.75 nm/mW. In this case, the rise and fall times were 1.11 $\mu$s and 1.47 $\mu$s, respectively, as shown in Figure 2b.

These studies underscore the substantial merits inherent in graphene-based methodologies relative to conventional microheater techniques within the realm of silicon photonics, particularly in terms of their superior tuning efficiency and remarkable modulation speed. Graphene's exceptional electrical and thermal properties, combined with its ultrahigh carrier mobility, confer a distinct advantage in precisely and rapidly modulating optical signals, offering a compelling alternative to traditional micro-heating methods. This technological shift holds profound implications for advancing the field of silicon photonics, enabling more efficient and faster modulation processes with potential applications spanning from telecommunications to data processing. Some different graphene-based thermos-optic modulators are demonstrated in Table 2 with their response time and the tuning efficiency parameters.

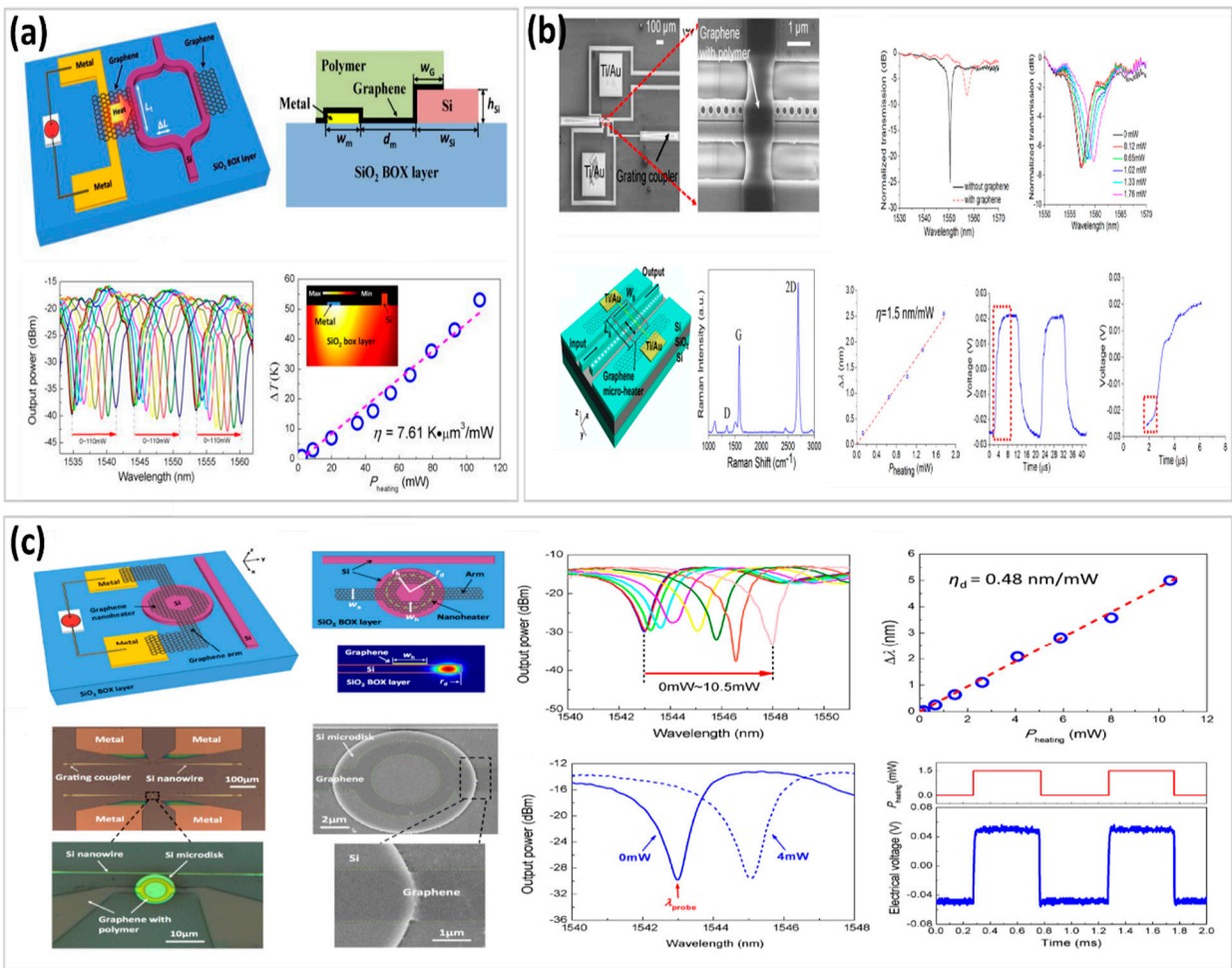

**Figure 2.** Graphene-assisted thermo-optic modulators. (**a**) Thermally tuning MZI with a non-local traditional metal heater and a graphene-based transparent flexible heat conductor [53], (**b**) Graphene-based silicon photonic crystal nanobeam cavity thermos-optic modulator [58], and (**c**) Thermally tunable silicon photonic microdisk resonator with a transparent graphene nanoheater [56].

**Table 2.** Performance summary of graphene-based thermo-optical modulators.

| Year | Structure | Response Time (μs) | Tuning Efficiency (nm/mW) | Spacer | Ref. |
|------|-----------|--------------------|---------------------------|--------|------|
| 2015 | Si MRR | 0.75 | 0.104 | No | [54] |
| 2016 | Si MRR | 3 | 0.33 | Yes | [55] |
| 2016 | Si Micro-disk | 12.8 | 1.67 | No | [56] |
| 2017 | Si PhC | 0.75 | 1.07 | No | [57] |
| 2017 | Si nanobeam | 1.47 | 1.5 | No | [58] |
| 2022 | Si RR-MRR | 2.4 | 0.24 | No | [59] |

## 3. High-Performance Graphene-Enabled Photodetectors

While an excellent waveguiding material, silicon inherently cannot be used to detect light in the telecommunication window of the electromagnetic spectrum. Therefore, researchers have investigated alternate photodetection materials to overcome this problem. Mainstream silicon photonics uses germanium (Ge) or monolayer graphene as photodetec-

tion materials. Germanium is a semiconductor with a high absorption coefficient in the C-band spectrum, making it ideal for photodetection [60–65].

Monolayer graphene, with its broad-spectrum absorption and tunability, is another attractive material for silicon photonic high-performance photodetectors. Graphene-enabled silicon photodetectors employ three distinct mechanisms—(a) Photovoltaic effect (PV), (b) Photo-bolometric effect (PB), and (c) Photo-thermoelectric effect (PTE) for photodetection, each offering unique advantages and characteristics.

### 3.1. Photovoltaic Effect (PV)-Based Photodetection

The first mechanism is the photovoltaic effect (PV), where incoming photons with energies exceeding the bandgap of silicon are absorbed to generate electron-hole pairs [60,61]. Xia et al. experimentally demonstrated a graphene-based photodetector up to 40 GHz in 2009 [66]. Later, Urich's team presented measurements of the intrinsic response time of metal–graphene–metal photodetectors with monolayer graphene using an optical correlation technique with ultrashort laser pulses. They obtained a response time of 2.1 ps, mainly given by the short lifetime of the photogenerated carriers with a bandwidth of ∼262 GHz. They also showed the dependence of the response time on gate voltage and illumination laser power (Figure 3a) [67]. In graphene-based photodetectors, when light is incident normally with respect to the graphene surface, the responsivity, or the ability of the device to convert light into an electrical signal, is often relatively low. This lower responsivity can be attributed to the limited interaction of photons with an atomically thin graphene layer. Additionally, graphene's inherently low absorption coefficient, especially in the visible and near-infrared spectrum, also contributes to this limitation.

To address this challenge and enhance the performance of graphene-based photodetectors, researchers have extensively explored the use of waveguide-based photodetectors [16,68–71]. Silicon waveguides are optical structures that can confine and guide light along a specific path, increasing the interaction length between light and the graphene layer. Integrating graphene onto the waveguide structures significantly improves the light–graphene interaction, leading to better photodetection performance [16]. These waveguide-based photodetectors are designed in such a way that incoming light is guided and channeled through the graphene layer for increased absorption. This extended interaction path allows for a higher probability of photon absorption by the graphene, resulting in improved responsivity and sensitivity of the photodetector. In this regard, Wang et al. showed that with a few layers of graphene on top of a silicon slot waveguide, a very strong absorption of light in graphene with 0.935 dB/um at the wavelength of 1.55 μm and a high responsivity of 0.273 A/W of the photodetector was achieved [63]. Schall et al. have demonstrated CVD-grown graphene on silicon waveguide-based photodetectors [72]. This device showed a maximum extrinsic response of 16 mA/W (46 mA/W absorption normalized) and a bandwidth of 41 GHz in the telecommunication band at 1.55 μm. This responsivity is approximately 80% higher than the conventional germanium-based photodetectors.

### 3.2. Photo-Bolometric Effect (PB)-Based Photodetection

In the photo-bolometric effect (PB), absorbed photons heat the graphene–silicon structure, changing its electrical resistance. This change in resistance is directly proportional to the incident light intensity and can be detected as a photocurrent. Plasmonic-based devices have also demonstrated superior responsivities. In hybrid plasmonic waveguides, the subwavelength confinement of the plasmonic modes gives additional advantages to enhance the light–graphene interaction. Due to the compatibility with traditional microelectronics fabrication methods, graphene-integrated hybrid plasmonic-waveguides photodetectors can also be fabricated on a large scale [69–71]. An integrated graphene photodetector of compact dimensions, featuring a graphene coverage length of 2.7 μm, was successfully presented by Ding et al. [69]. This photodetector exhibited an impressive unaltered response characteristic even at exceedingly high frequencies up to 110 GHz. This remarkable performance is accompanied by an intrinsic responsivity reaching 25 mA/W, signifying its

exceptional sensitivity to incident light. Scaling up the device dimensions to 19 μm substantially enhanced the intrinsic responsivity to 360 mA/W. As the gap between the plasmonic slots becomes less than 50 nm, the photovoltaic (PV) mechanism starts converting into a photo-bolometric effect (PB) mechanism within the photodetection process, and this phenomenon was experimentally presented by Ma et al. (2019), as shown in Figure 3b [73]. In PB, the absorbed photons heat the graphene–silicon structure, changing its electrical resistance. This change in resistance is directly proportional to the incident light intensity and can be detected as a photocurrent. Zhizhen Ma et al. (2020) proposed a very compact plasmonic slot graphene-based photodetector with a high responsivity of 0.7 A/W [74]. In this device, they realized a 30-times-higher responsivity with a 15 nm narrower plasmonic slot, as shown in Figure 3d. The detailed discussion of graphene photodetectors based on these three mechanisms provides insights into their operation, performance, and potential applications in various photonic and optoelectronic systems by using the high absorption capability of graphene in the C-band.

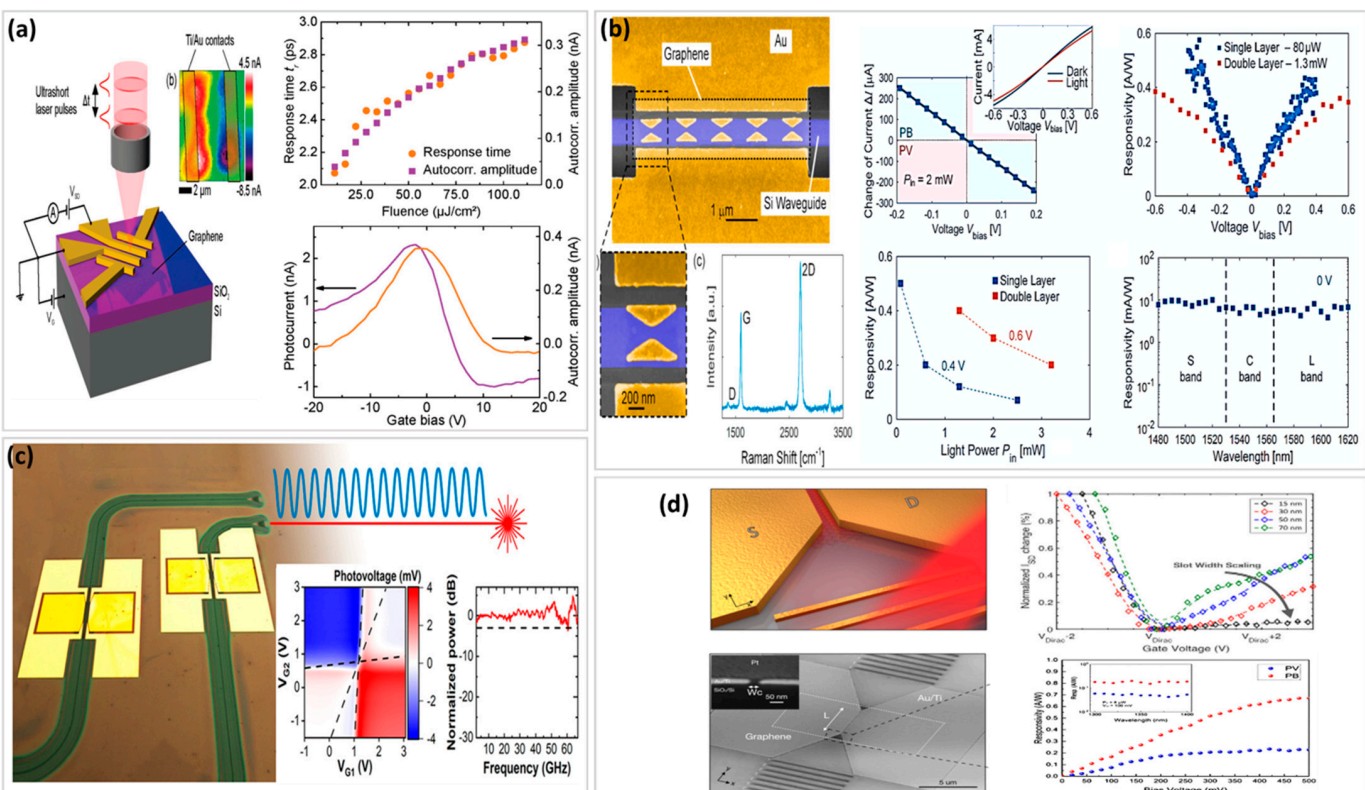

**Figure 3.** Graphene-assisted photodetectors. (**a**) Metal–graphene–metal photodetector with monolayer graphene [67]. (**b**) Plasmonically enhanced graphene photodetector [73]. (**c**) Graphene photodetector with polymeric gate dielectric on passive photonic waveguide [75], and (**d**) Compact graphene plasmonic slot photodetector on silicon-on-insulator [74].

### 3.3. Photo-Thermoelectric Effect (PTE) Based Photodetection

The photo-thermoelectric effect (PTE) in photodetectors relies on the generation of a temperature gradient within the device due to absorbed photons. This temperature gradient induces a voltage across the device, which can be measured as a photocurrent. PTE-based photodetectors are particularly efficient at detecting low-energy photons. This phenomenon is also used in the graphene-enabled silicon photodetector with the support of different structures, including MRRs) [76], photonic crystal waveguides [77], and double-layer graphene [75] to increase the performance of the device. The performance analysis of a standard silicon photodetector enhanced with graphene and cutting-edge photodetectors utilizing traditional bulk materials is presented in Table 3 for comparison. It can be seen

from Table 3 that only silicon waveguide structures have limited bandwidth with limited responsivity. But, when graphene was used with the plasmonics-based photodetectors, the bandwidth was more than 110 GHz with a responsivity of 0.36 A/W. Such structures find huge promise in on-chip silicon-integrated photonic platforms. The performances of typical conventional photodetectors based on different bulk materials for comparison with state-of-art graphene based photodetectors is presented in Table 4.

**Table 3.** Performance summary of graphene-based photodetectors.

| Year | Structure | Wavelength [μm] | Bandwidth [GHz] | Responsivity (A/W) | Response Time [ps] | Ref. |
|------|-----------|-----------------|-----------------|--------------------|--------------------|------|
| 2013 | Si waveguide | 2.75 | - | 0.13 | - | [78] |
| 2013 | Silicon waveguide | 1.31–1.65 | 18 | 0.03 | 25 | [79] |
| 2015 | Silicon waveguide | 1.55 | 42 | 0.36 | 3 | [80] |
| 2016 | Plasmonic waveguide | 1.55 1.55 | - - | 0.085 0.37 | - - | [81] |
| 2019 | Hybrid plasmonic waveguide | 1.48–1.62 | >110 | 0.5 | - | [73] |
| 2020 | Plasmonic waveguide | 1.480–1.62 | >110 | 0.36 | - | [69] |
| 2020 | Hybrid plasmonic Si waveguide | 1.55 and 2.00 | >40 | 0.4 | - | [82] |
| 2021 | Si double slot waveguide | 1.55 | 78 | 0.6 | - | [71] |

**Table 4.** Performance summary of conventional photodetectors.

| Year | Absorption Material | Wavelength [μm] | Bandwidth [GHz] | Responsivity (A/W) | Ref. |
|------|---------------------|-----------------|-----------------|--------------------|------|
| 2021 | Ge | 1.55 | 265 | 0.3 | [62] |
| 2018 | $\alpha$-Ge | 1.27–1.33 | >100 | 0.35 | [83] |
| 2015 | InP | 1.31–1.55 | 130 | 0.5 | [84] |
| 2021 | InP | 1.24–1.65 | 40 | 0.8 | [85] |
| 2016 | InGaAs | 1.26–1.36 | 32 | 0.68 | [86] |

## 4. Future Directions and Challenges

As the above literature reveals, significant developments have been made in graphene-based modulators and photodetectors. The previously discussed studies together indicate that graphene has significant potential as a material for the development of optical modulators and photodetectors that provide high-speed capabilities and low power consumption [87,88]. Meanwhile, to obtain ultra-fast speed modulators, series resistors should be decreased by optimizing the fabrication process. In this way, chip-integrated high-speed EO modulators with a modulation speed of >100 GHz [88,89] and a power consumption of <1 fJ/bit could be obtained [90]. Moreover, graphene is also an ideal material for free-space light modulation. Inspired by the demonstration in the mid-infrared, high-speed near-infrared free space optical modulators could also be realized by using novel resonators and graphene. Such GHz free-space modulators could be very useful in spatial light communication, beam steering, optical communication, etc.

In addition, future challenges for high-performance graphene-based optical modulators and photodetectors may lie in practical issues in mass production [91]. So far, the majority of investigations have relied on the wet transfer technique for depositing chemical vapor deposition (CVD)-grown or exfoliated graphene onto the desired substrate. The current state of the constructed devices exhibits a lack of the usual homogeneity, though they remain adequate for prototype demonstrations. In addition, the wet transfer procedure has the potential to introduce particles that might greatly amplify the transmission loss of the silicon chip. Another challenge is to increase the light–graphene interaction. While the atomic thickness of graphene has proven advantageous in some applications, such as transparent microheaters, it also presents limitations in terms of poor modulation depth and responsivity in photodetectors. Considerable work has been carried out to improve the absorption capabilities of the graphene layer, resulting in noteworthy advancements. Nevertheless, the current performance in relation to the modulation depth and responsivity is comparable only to devices using bulk materials, although at the cost of compromising bandwidth. Hence, the pursuit of improving the interaction between light and graphene while also preserving the benefits of graphene-based devices is a formidable but very significant subject of study.

## 5. Conclusion and Perspectives

Over the course of the previous decade, there has been a notable and remarkable emergence of graphene in the field of silicon photonics. Several research groups have reported the development of silicon photonic modulators and photodetectors that exhibit amazing performance by utilizing the remarkable features of graphene. In general, the fields of silicon photonics and graphene technology are currently undergoing significant advancements, with ongoing efforts to further their development. Moreover, the combination of these two innovative platforms in a hybrid manner exhibits substantial promise. It is posited that upon the comprehensive resolution of the aforementioned obstacles, hybrid silicon/graphene optoelectronic devices will engender novel technological capacities, hence benefiting both industrial and scientific domains. It is obvious that with the tremendous growth of material engineering, different 2D materials with different band gaps will also play an important role in increasing the performance of silicon-based modulators and photodetectors. Overall, there is a gap for further development in the field of graphene-based silicon photonics and their applications beyond the above review applications, as discussed in the above section (Future directions and challenges). For the future, graphene-enabled silicon-based devices hold significant potential for advancing various fields.

In general, the fields of silicon photonics and graphene technology are now undergoing significant advancements, with ongoing efforts towards their fast growth. Moreover, the combination of these two innovative platforms in a hybrid manner exhibits considerable promise. It is suggested that with the comprehensive resolution of the mentioned challenges, the integration of silicon and graphene in optoelectronic devices would facilitate the emergence of novel technological capacities, therefore benefiting both industrial and scientific domains. This shift is likely to have transformational effects on data transmission, sensing, and a wide range of other applications that rely on photonics technology.

**Funding:** This research received no external funding.

**Institutional Review Board Statement:** Not applicable.

**Informed Consent Statement:** Not applicable.

**Data Availability Statement:** Not applicable.

**Acknowledgments:** Some of the work cited is based on research sponsored by Air Force Research Laboratory (AFRL) under Agreement Number FA8650-20-2-5506. The U.S. Government is authorized to reproduce and distribute reprints for Governmental Purposes notwithstanding any copyright notation thereon.

**Conflicts of Interest:** The authors declare no conflict of interest.

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
