# Peer review of "Recent Advances in Graphene-Enabled Silicon-Based High-Speed Optoelectronic Devices—A Review"

_photonics, doi:10.3390/photonics10121292_

Round 1

Reviewer 1 Report

Comments and Suggestions for Authors

REVIEW

on the manuscript “Recent Advances in Graphene Enabled Silicon based High-Speed Optoelectronic Devices A Review

by Yadvendra Singh and Harish Subbaraman

In the presented manuscript the review of the recent results in the experimental investigations of using graphene in silicon-based optoelectronic devices are presented. Brief tutorial-like introduction into the basic parameters of the discussed materials and technology is given. Examples of various types of graphene-enabled modulators (including electro-optic and thermo-optic modulation devices) and photodetectors (including photovoltaic, photo-bolometric and photo-thermoelectric devices) based on silicon compatible technology are considered. Their principles of operation, functional parameters and characteristics are discussed. Comparison tables for the figures of merit for various types of structures for the same material are presented.

Overall, the paper is organized and written well and at high scientific level.

However, in the reviewer’s opinion, the manuscript should be slightly improved before publication. The details are listed below.

1. The main question about this review is why the perspectives and challenges of these materials system in future technologies are not discussed. These may be done in the separate section.

2. Comparison of performance characteristics with other types of Si-compatible photodetectors should be added.

3. Some up-to-date references should be added, such as

Recent Advances in Si-Compatible Nanostructured Photodetectors // Technologies, 2023. (For comparison with Ge/Si and GeSn/Si devices)

Zero Bias Operation: Photodetection Behaviors Obtained by Emerging Materials and Device Structures // Micromachines, 2022. (For comparison with other types of zero-bias devices)

4. Page 5: Different abbreviations for ‘micro’ prefix are used inconsistently throughout the text.

5. Page 5: MDR and MRR abbreviations should be deciphered.

6. One other recommendation about this manuscript is to double-check the reference in Figure 1a, because the reference in the text ([30]) is not consistent with reference in the figure caption ([32]) (Pages 2 and 4).

7. Sub-section “Photo-thermoelectric effect (PTE) based photodetection” should be (c) instead of (a).

Conclusion: The presented manuscript may be published in the Photonics journal after moderate revision.

Comments on the Quality of English Language

English language is fine. Some minor formatting is needed.

Reviewer 2 Report

Comments and Suggestions for Authors

This manuscript reviews the development of graphene enabled-silicon based optoelectronic devices. Overall, the manuscript is well written. I would like to recommend it to be published in Photonics if the authors can address following minor comments:

1.     In addition to in-depth research on graphene-silicon based photodetectors and modulators, graphene-silicon based optoelectronic devices have also been applied in various fields such as phase shifters and nonlinear optical signal processing, please supplement the relevant content.

2.     Authors should make appropriate extensions (or outlooks) in the conclusion section.

3.     In Tables 1, 2, and 3, the authors should supplement and compare relevant studies in the past year and two years.

4.     There are some highly related recent references for graphene-based optoelectronic devices missing, for example, [Journal of the Optical Society of America B 38 (10), 3206-3211 (2021)], [Journal of the Optical Society of America B 40 (2), 233-238 (2023)], [Optics Express 27 (4), 5253-5263 (2019)].

Reviewer 3 Report

Comments and Suggestions for Authors

This review paper summarized the recent advances of the graphene enabled silicon based optoelectronics. The author very clearly described the advantages brought by graphene when using it for different kinds of silicon based optoelectronics, for example, the electro-optic modulator, thermo-optic modulator and various types of photo-detectors. This is a very informative review that can help readers quickly understand the current progress in this field. It would be better to include some discussion on the future directions of research on the graphene applications on optoelectronics.

Below are some detailed comments:

1) Page 4, section b. The author mentioned ‘Its unique ability to absorb light efficiently across a wide range of spectrum make it stand out’. Could the author elaborate a bit more on why this property will be beneficial for the thermo-optic modulator?

2) Page 5. I recommend adding some introduction to the performance of conventional thermo-optic modulators, i.e. modulation depth, response time etc. to help readers quickly get an idea how much improvement is introduced by the graphene.

3) Page 7. The author mentioned that the waveguide can improve the responsivity of the graphene. However, the waveguide is sensitive to the wavelength. Could author comments on whether it is feasible to apply graphene for photo-detection for broadband visible range? Like a conventional silicon based detector with good responsivity from 300 - 1100 nm.

4) Author mentioned ‘there is a gap for further development in the field of graphene-based silicone photonics ....’. Could you include what this gap is?

Comments on the Quality of English Language

Very good.

In ‘Conclusion’, row 6: ‘with ongoing efforts to further their development’, recommend change to ‘with ongoing efforts for their further development’.
